# Equivalent Static Wind Load for Structures with Inerter-Based Vibration Absorbers

Ning Su [1] 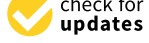, Shitao Peng [1,*], Zhaoqing Chen [2], Ningning Hong [1] and Yasushi Uematsu [3]

[1] Tianjin Research Institute for Water Transport Engineering, Ministry of Transport of People's Republic of China, Tianjin 300456, China
[2] School of Civil Engineering and Architecture, Northeast Electric Power University, Jilin 132012, China
[3] National Institute of Technology (KOSEN), Akita College, Akita 011-8511, Japan
* Correspondence: pengshitao@tiwte.ac.cn

**Abstract:** Equivalent Static Wind Loads (ESWL) are desired in structural design to consider peak dynamic wind effects. Conventional ESWLs are for structures without control. For flexible structures with vibration control devices, the investigation of ESWL is required. Inerter-based Vibration Absorbers (IVAs), due to the light weight and high performance, gained much research attention recently. This paper established a generic analytical framework of ESWL for structures with IVAs. The analytical optimal design formulas for IVAs with different configurations and installation locations are provided. Subsequently, the solutions to uncontrolled and controlled wind-induced responses are derived based on the filter approach. Finally, the ESWL for controlled structures are presented with a gust response factor approach. The ESWL estimation for a tall chimney controlled by IVAs is illustrated, and the results revealed a significant ESWL reduction effect of the IVAs, particularly for the cross-wind vortex resonance. In the presented framework, the conventional uncontrolled ESWL can be converted to the controlled one with a control ratio. The closed form solution of the control ratio is provided, which enables a quick estimation of ESWL for controlled structures particularly in the preliminary design stage. The presented approach has the potential to be extended to more complex structures and vibration control devices.

**Keywords:** equivalent static wind load; dynamic vibration absorber; gust response factor; wind-induced response; vibration control; inerter



## 1. Introduction

The Equivalent Static Wind Load (ESWL) is an important concept in structural wind resistance designs. It provides a simplified procedure to estimate the peak dynamic wind effect on structures, which can be combined with other load effects in structural design. The basic framework of ESWL was proposed as gust loading factors by Davenport [1]. The ESWL concept and method have been extensively developed and extended considering more complex situations [2–15]. These studies provide a basis for international codes for wind loading [16,17]. However, these ESWL approaches are for structures without vibration control.

The development on material and construction technologies enables structures to become more flexible. Vibration control devices are commonly applied to modern structures. Tuned Mass Damper (TMD) is one of the most classical Dynamic Vibration Absorbers (DVAs) for reducing the vibration responses of flexible structures. The vibration control performances are strongly dependent on the tuning parameters. Den Hartog [18] proposed a theoretic framework to obtain the optimal parameters analytically, which aims at minimizing the norms of the dynamic amplification functions. Based on the theory, the analytical optimal parameters of TMD for harmonic and white-noised excitations are derived [19]. According to this criterion, the control performance of the TMD is highly dependent on the tuning mass ratio. Achieving a lightweight design of the DVA is a challenging task.

In order to further enhance the vibration control performance, many investigations were performed. The inerter device was invented for both mechanical and civil engineering fields [20,21]. It can produce a control force in proportion to the relative acceleration at its two ends implemented by proper mechanical configurations, such as a rack-pinon-flywheel, screw-ball systems, etc. [22]. Thus, an inertial effect of thousands of times its physical mass can be provided by an inerter. The Inerter-based Vibration Absorbers (IVAs) have the potential to achieve lightweight and high performance vibration control, which has recently attracted much attention in research.

The first IVA realized in practical civil engineering structures is the Tuned Mass Viscous Damper (TVMD) [21], which is installed on a tall building located in Sendai, Japan. Replacing the dashpot of the TMD with TVMD, a Rotational inertia double-tuned mass damper (RIDTMD) [23] was proposed, which significantly reduces the mass of the DVA [24]. Other than TVMD, many other IVA configurations are proposed and investigated. The Tuned Mass Damper Inerter (TMDI) was proposed by adding an inerter between the mass block of the TMD and the ground [25], which is beneficial in base-isolation system [26,27]. Considering the practical installation, the connection of the inerter of TMDI was moved from the ground to the structure body [28,29] to form an inter-layer device, or adjacent buildings, forming a damped link [30,31]. Replacing the mass of the TMD with an inerter, Tuned Inerter Dampers (TIDs) were implemented and investigated [32,33]. A variant design of the above mentioned DVA was achieved to enhance the control performance by connecting the dashpot to the other side. Such as a variant design of TMD (VTMD, [34,35]), a variant design of TMDI (VTMDI, [28,36]) is similar. It is interesting to find that the TVMD and TID forms a variant design pair, and they are extensively compared with each other in the literature [37–39]. More recently, the above DVAs are investigated considering nonlinearity effects, such as [40,41].

Although both the IVAs and ESWLs are extensively addressed, they are investigated separately. Because the practical application of IVAs becomes common, it is hoped to develop the corresponding ESWLs for controlled structures. In order to address this gap, the present paper aims at establishing a generic analytical framework of ESWL for structures with IVAs. The analytical optimal design formulas for IVAs with different configurations and installation locations are provided. Subsequently, the solutions to uncontrolled and controlled wind-induced responses are derived based on the filter approach [42]. Finally, the ESWLs for controlled structures are presented with a gust response factor approach. As an example of the application, the ESWL estimation for a tall chimney controlled by IVAs is performed at the end of the paper.

## 2. Inerter-Based Vibration Absorbers

The equivalent static wind load (ESWL) is based on the wind-induced peak response. For the structures coupling with inerter-based vibration absorbers (IVAs), the wind-induced responses are controlled. Moreover, the control performance of the IVA is highly dependent on the tuning parameters. Therefore, it is the basic task to establish the equations of motion, and determine the optimal parameters and performances of IVA. In this section, a variety of IVAs are formulated generically. The optimal parameters are analytically obtained with the Fixed-point approach.

### 2.1. Generic Equations of Motion

In order to simplify the derivation, the primary structure is assumed to be a generalized single-degree-of-freedom (SDOF) structure. Assuming that the fundamental mode dominates a slender structure as an example, the equation of motion is described with the generalized mass, stiffness, and damping, denoted as $M$, $K$, and $C$, respectively. It is dynamically characterized by the natural circular frequency $\omega_n = \sqrt{K/M}$, and damping ratio $\zeta_n = C/2\sqrt{KM}$. As shown in Figure 1, the displacement response time–history $u(z, t)$ can be decoupled as a spatial variant normalized modal function $\Phi(z)$, with respect to the generalized coordinate (height) $z$ ($0 \leq z \leq H$, $\Phi(H) = 1$), and a time dependent generalized

displacement response $x(t)$, i.e., $u(z, t) = \Phi(z) \cdot x(t)$. Note that $x(t)$ is exactly equalled to the top displacement, $x(t) = u(H, t)$.

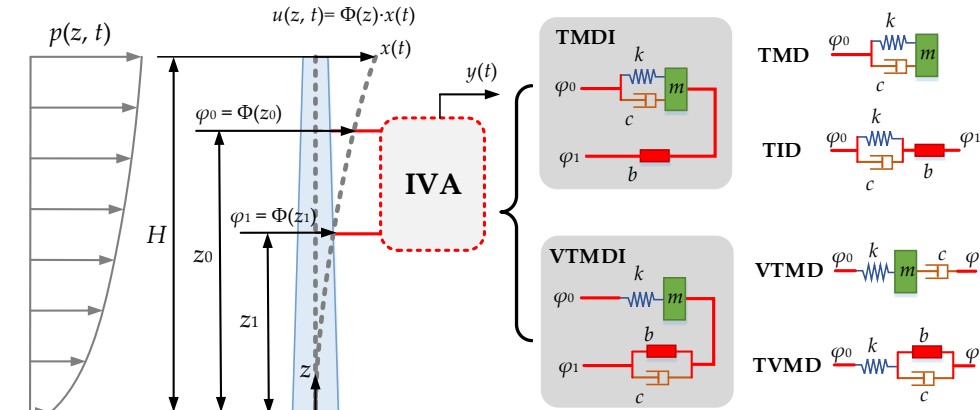

**Figure 1.** A generalized SDOF system coupling with an IVA of various configurations.

For an uncontrolled primary structure subjected to wind load $p(z,t)$, the vibration is governed by Equation (1). In the equation, $F(t) = \int_0^H p(z, t)\Phi(z)\mathrm{d}z$ is the generalized wind load. The wind load is composed of static and dynamic components, statistically denoted by the mean value $\overline{F}$ and standard deviation value $\sigma_F$. The normalized transfer function $H(s)$ based on the Laplace transform of Equation (1) is written as Equation (2), with $s$ being the complex frequency, and $X(s)$ and $F(s)$ being the Laplace transform of the output response $x(t)$ and the input excitation $F(t)$, respectively. When $s$ takes the pure imaginary frequency $i\omega$, $H(i\omega)$ is the normalized frequency response function. $V(s) = (s/\omega_n)^2 + 2\zeta_n s/\omega_n + 1$ is the inverse of $H(s)$. Here, $i = \sqrt{-1}$ is an imaginary unit. Consequently, the static and dynamic wind-induced responses, denoted by the mean and standard deviation values, $\overline{x}$ and $\sigma_x$, are shown in Equation (3). The peak response $\hat{x}$ is obtained from the peak factor approach, as shown in Equation (3). According to Davenport's statistical approach [1], the peak factor $g$ is approximated with $\sqrt{2 \ln n_0 T} + \frac{\gamma}{\sqrt{2 \ln n_0 T}}$, where $T$ is the time duration, $n_0$ is the mean up-crossing rate that can be approximated by $\omega_n/2\pi$, and $\gamma$ is the Euler constant taken as 0.5772.

$$M\ddot{x} + C\dot{x} + Kx = F(t) \tag{1}$$

$$H(s) = \frac{X(s)}{F(s)/K} = \frac{K}{Ms^2 + Cs + K} = \frac{1}{(s/\omega_n)^2 + 2\zeta_n s/\omega_n + 1} = \frac{1}{V(s)} \tag{2}$$

$$\begin{cases} \overline{x} = H(0) \cdot \overline{F}/K \\ \sigma_x = \frac{\sigma_F}{K}\sqrt{\int_0^\infty S_F(\omega)|H(i\omega)|^2 \mathrm{d}\omega} \\ \hat{x} = \overline{x} + g\sigma_x \end{cases} \tag{3}$$

In this paper, two major configurations of IVAs, TMDI, and VTMDI are considered, as shown in Figure 1. It is also noted that TMD and TID can be expressed as TMDI with an absent of mass. Likewise, VTMD and TVMD can be VTMDI with an absent of mass. Therefore, these configurations are the major IVAs with similar components and different configurations. They can be formulated and modeled generically. Although the IVAs are discussed in several papers [37–39], the ESWL of structures with these IVAs is not fully addressed.

For a primary structure controlled by an IVA installed between coordinates $z_0$ and $z_1$, as shown in Figure 1, the Ritz-Galerkin method is adopted, as referred to [29,39,43,44], assuming that $u(z, t) = \Phi(z) \cdot x(t)$. According to the principle of visual work, the equations of motion are rewritten as Equation (4). In the equation, $\varphi_0 = \Phi(z_0)$ and $\varphi_1 = \Phi(z_1)$ are the location parameters of the IVA. $f_0$ and $f_1$ are the control force generated by the IVA

at installation locations $z_0$ and $z_1$, respectively. $y$ represents the absolute displacement of the IVA.

$$\begin{cases} M\ddot{x} + C\dot{x} + Kx - (\varphi_0 f_0 + \varphi_1 f_1) = F(t) \\ m\ddot{y} + f_0 + f_1 = 0 \end{cases} \tag{4}$$

Considering a Tuned Mass Damper Inerter (TMDI) with mass $m$, stiffness $k$, damping $c$, and inertance coefficients $b$ as an example, $f_0$ and $f_1$ are written in Equation (5). Note that when $b = 0$, it become a Tuned Mass Damper (TMD). Whereas, when the mass can be ignored ($m = 0$), it has a similar configuration with a Tuned Inerter Damper (TID). Thus, Equation (5) is applicable for all of the above-mentioned situations. If the dashpot of the TMDI is connected to the inerter side, a variant design of TMDI is formed, namely VTMDI. In this case, $f_0$ and $f_1$ are expressed as Equation (6). When the inerter is absent, it forms a variant design of TMD (VTMD). When the mass becomes absent, it has a similar configuration with the Tuned Viscous Mass Damper (TVMD [21]), which is also denoted as TID2 in [37]. Equation (6) is applicable for these variants. Also note that, when $\varphi_1 = 0$, the IVAs are connected to the ground, known as grounded IVAs. Conventional IVAs usually takes $\varphi_0 = 1$ and $\varphi_1 = 0$ regardless of the installation locations. They are assumed to be connected between the tip of the building and the ground.

$$\begin{cases} f_0 = k(y - \varphi_0 x) + c(\dot{y} - \varphi_0 \dot{x}) \\ f_1 = b(\ddot{y} - \varphi_1 \ddot{x}) \end{cases} \tag{5}$$

$$\begin{cases} f_0 = k(y - \varphi_0 x) \\ f_1 = b(\ddot{y} - \varphi_1 \ddot{x}) + c(\dot{y} - \varphi_1 \dot{x}) \end{cases} \tag{6}$$

Substituting Equation (5) or Equation (6) into Equation (4), we can obtain the normalized transfer function $H(s)$. $H(s)$ can be formatted as a rational expression, as Equation (7). The denominator polynomial $\Gamma(s) = \sum\limits_{j=0}^{4} \gamma_j (s/\omega_\mathrm{n})^j$ is quartic, with dimensionless coefficients $\gamma_j$ ($j = 0, 1, 2, 3, 4$). The numerator polynomial $\Theta(s) = \sum\limits_{j=0}^{2} \theta_j (s/\omega_\mathrm{n})^j$ is quadratic, with dimensionless coefficients $\theta_j$ ($j = 0, 1, 2$).

$$H(s) = \frac{X(s)}{F(s)/K} = \frac{\Theta(s)}{\Gamma(s)} = \frac{\sum\limits_{j=0}^{2} \theta_j (s/\omega_\mathrm{n})^j}{\sum\limits_{j=0}^{4} \gamma_j (s/\omega_\mathrm{n})^j} \tag{7}$$

In order to express the equations in a dimensionless form, the tuning parameters of the IVA are defined, as shown in Table 1.

**Table 1.** Definitions of the tuning parameters of the IVA.

| Tuning Parameter | Symbol | Definition |
|---|---|---|
| Tuning mass ratio | $\mu$ | $m/M$ |
| Tuning inertance ratio | $\beta$ | $b/M$ |
| Nominal frequency | $\omega_\mathrm{d}$ | $\sqrt{k/(m+b)}$ |
| Nominal damping ratio | $\nu$ | $c/2\sqrt{k(m+b)}$ |
| Tuning frequency ratio | $\zeta_\mathrm{d}$ | $\omega_\mathrm{d}/\omega_\mathrm{n}$ |

For TMDI and VTMDI, the coefficients are expressed, as shown in Table 2. Consequently, the controlled wind-induced responses can be calculated by Equation (3), merely adopting the normalized transfer function $H(s)$ as Equation (7).

**Table 2.** Dimensionless coefficients of the normalized transfer function $H(s)$ for TMDI and VTMDI.

| Coefficient | TMDI | VTMDI |
|---|---|---|
| $\gamma_0$ | $\nu^2$ | $\nu^2$ |
| $\gamma_1$ | $2\zeta_n\nu^2 + 2\zeta_d\nu$ | $2\zeta_n\nu^2 + 2\zeta_d\nu\left[1 + \nu^2(\mu+\beta)(\varphi_0 - \varphi_1)^2\right]$ |
| $\gamma_2$ | $1 + 4\zeta_n\zeta_d\nu + \nu^2\left[1 + \mu\varphi_0^2 + \beta(\varphi_0 - \varphi_1)^2\right]$ | $1 + 4\zeta_n\zeta_d\nu + \nu^2\left[1 + \mu\varphi_0^2 + \beta(\varphi_0 - \varphi_1)^2\right]$ |
| $\gamma_3$ | $2\zeta_n + 2\zeta_d\nu\left[1 + \mu\varphi_0^2 + \beta(\varphi_0 - \varphi_1)^2\right]$ | $2\zeta_n + 2\zeta_d\nu(1 + \mu\varphi_1^2)$ |
| $\gamma_4$ | $1 + \frac{\mu}{\mu+\beta}\beta\varphi_1^2$ | $1 + \frac{\mu}{\mu+\beta}\beta\varphi_1^2$ |
| $\widetilde{\gamma}_1$ | $2\nu$ | $2\nu\left[1 + \nu^2(\mu+\beta)(\varphi_0 - \varphi_1)^2\right]$ |
| $\widetilde{\gamma}_2$ | $1 + \nu^2\left[1 + \mu\varphi_0^2 + \beta(\varphi_0 - \varphi_1)^2\right]$ | $1 + \nu^2\left[1 + \mu\varphi_0^2 + \beta(\varphi_0 - \varphi_1)^2\right]$ |
| $\widetilde{\gamma}_3$ | $2\nu\left[1 + \mu\varphi_0^2 + \beta(\varphi_0 - \varphi_1)^2\right]$ | $2\nu(1 + \mu\varphi_1^2)$ |
| $\theta_0$ | $\nu^2$ | $\nu^2$ |
| $\theta_1$ | $2\zeta_d\nu$ | $2\zeta_d\nu$ |
| $\theta_2$ | $1$ | $1$ |
| $\widetilde{\theta}_1$ | $2\nu$ | $2\nu$ |

### 2.2. Analytical Optimal Design Based on Fixed-Point Approach

The next step is to determine the appropriate parameters of the IVA. Basically, the optimal design is to determine the optimal tuning parameters $\{\nu_{\text{opt}}, \zeta_{\text{dopt}}\}$ with respect to the input parameters $\{\mu, \beta, \varphi_0, \varphi_1\}$. The optimization can be performed based on different performance targets, e.g., the maximum (infinity norm) or 2nd norm of the dynamic amplification function ($H_\infty$, $H_2$ optimization). Among them, the most basic optimal design method is the Fixed-point approach (FPA), which can lead to analytical results.

According to Den Hartog [18], the dynamic amplification function (DAF) $D(\lambda)$ is defined as the modulus of $H(\mathrm{i}\omega)$, particularly ignoring $\zeta_n$, as shown in Equation (8), where $\lambda = \omega/\omega_n$ is the normalized frequency. The polynomials $A_1(\lambda)$, $A_2(\lambda)$, $B_1(\lambda)$, and $B_2(\lambda)$ are determined by substituting Equation (7) into Equation (8), as shown in Equation (9). In the Equation, $\widetilde{\gamma}_1 = \frac{\gamma_1|_{\zeta_n=0}}{\zeta_d}$, $\widetilde{\gamma}_2 = \gamma_2|_{\zeta_n=0}$, $\widetilde{\gamma}_3 = \frac{\gamma_3|_{\zeta_n=0}}{\zeta_d}$, $\widetilde{\theta}_1 = \theta_1/\zeta_d$. For TMDI and VTMDI, the coefficients are shown in Table 2.

$$D(\lambda) = |H(\mathrm{i}\omega)|_{\zeta_n=0} = \sqrt{\frac{A_1^2(\lambda) + B_1^2(\lambda)\zeta_d^2}{A_2^2(\lambda) + B_2^2(\lambda)\zeta_d^2}} \tag{8}$$

$$\begin{cases} A_1(\lambda) = -\theta_2\lambda^2 + \theta_0 \\ A_2(\lambda) = \gamma_4\lambda^4 - \widetilde{\gamma}_2\lambda^2 + \gamma_0 \\ B_1(\lambda) = \widetilde{\theta}_1\lambda \\ B_2(\lambda) = (-\widetilde{\gamma}_3\lambda^2 + \widetilde{\gamma}_1)\lambda \end{cases} \tag{9}$$

Based on the FPA, for a determined tuning frequency ratio $\nu$, $D(\lambda)$ always passes through two fixed points, as shown in Figure 2a. The horizontal coordinates of the fixed points $\lambda_{1,2}$ can be solved by letting $\zeta_d$ be 0 and infinity, i.e., $\frac{A_1(\lambda)}{A_2(\lambda)} = \pm\frac{B_1(\lambda)}{B_2(\lambda)}$. At the optimal tuning frequency ratio $\nu_\infty$, the DAF values of the two fixed points are equal, i.e., $D(\lambda_1) = D(\lambda_2) = D_{\text{opt}}$. Consequently, we obtain the following equation:

$$\frac{A_1(\lambda_1)}{A_2(\lambda_1)} = \frac{B_1(\lambda_1)}{B_2(\lambda_1)} = -\frac{B_1(\lambda_2)}{B_2(\lambda_2)} \tag{10}$$

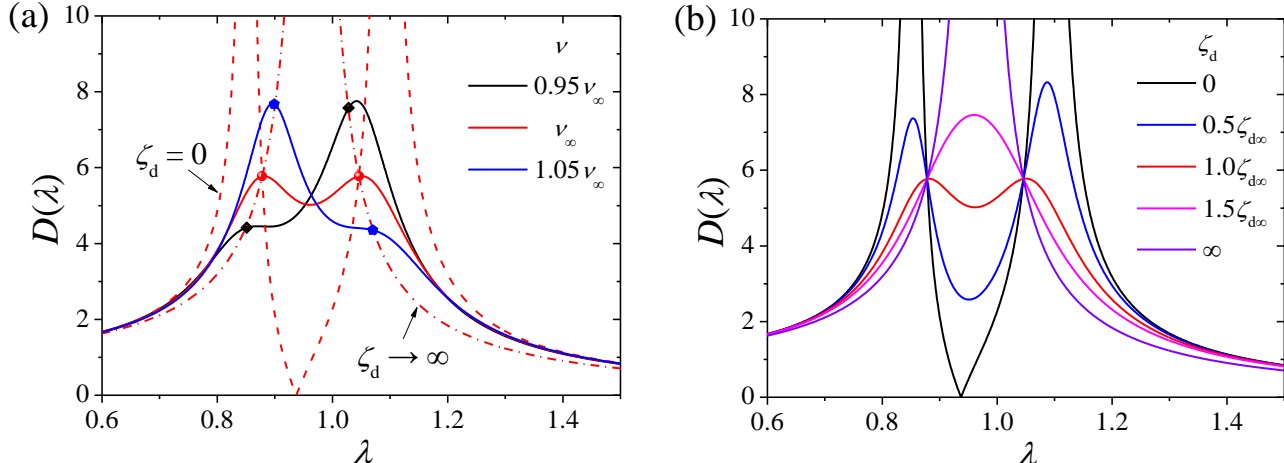

**Figure 2.** The DAF curves $D(\lambda)$ with different $\nu$ and $\zeta_d$. (**a**) $D(\lambda)$ with different $\nu$; the fixed points are equal when $\nu = \nu_\infty$. (**b**) $D(\lambda)$ at $\nu_\infty$ with different $\zeta_d$; the fixed points are approximately assigned to the equal peaks when $\zeta_d = \zeta_{d\infty}$.

By substituting Equation (9) into Equation (10), $\nu_\infty$ can be obtained by solving Equation (11) analytically. The coordinates of the optimal fixed points $(\lambda_{1,2}, D_{opt})$ are given by Equation (12).

$$\widetilde{\gamma}_3(\widetilde{\gamma}_3\theta_0 + \widetilde{\gamma}_1\theta_2 + \widetilde{\gamma}_2\widetilde{\theta}_1) - 2\widetilde{\gamma}_1(\widetilde{\gamma}_3\theta_2 + \gamma_4\widetilde{\theta}_1) = 0 \tag{11}$$

$$\begin{cases} D_{opt} = \widetilde{\theta}_1\sqrt{\dfrac{\widetilde{\gamma}_3\theta_2 + \gamma_4\widetilde{\theta}_1}{\widetilde{\gamma}_1^2(\widetilde{\gamma}_3\theta_2 + \gamma_4\widetilde{\theta}_1) - \widetilde{\gamma}_3^2(\widetilde{\gamma}_1\theta_0 + \gamma_0\widetilde{\theta}_1)}} \\ \lambda_{1,2} = \sqrt{\dfrac{\widetilde{\gamma}_1 \pm \widetilde{\theta}_1/D_{opt}}{\widetilde{\gamma}_3}} \end{cases} \tag{12}$$

According to FPA, the optimal tuning damping ratio $\zeta_{d\infty}$ can be determined by assigning the fixed points to the peaks of the DAF, i.e., $\frac{dD_{\zeta_d=\zeta_{d\infty}}^2(\lambda_{1,2})}{d\lambda} = 0$, as shown in Figure 2b. Based on the quotient rule of derivation, the following equation may be obtained:

$$\frac{\frac{dA_1^2(\lambda_{1,2})}{d\lambda} + \frac{dB_1^2(\lambda_{1,2})}{d\lambda}\zeta_{d\infty}^2}{\frac{dA_2^2(\lambda_{1,2})}{d\lambda} + \frac{dB_2^2(\lambda_{1,2})}{d\lambda}\zeta_{d\infty}^2} = \frac{A_1^2(\lambda_{1,2}) + B_1^2(\lambda_{1,2})\zeta_{d\infty}^2}{A_2^2(\lambda_{1,2}) + B_2^2(\lambda_{1,2})\zeta_{d\infty}^2} = D_{opt}^2 \tag{13}$$

Thus, the optimal tuning damping ratio $\zeta_{d\infty 1,2}$, corresponding to the fixed points at $\lambda_1$ and $\lambda_2$, are given by Equation (14).

$$\zeta_{d\infty 1,2} = \sqrt{\frac{D_{opt}^2\frac{dA_2^2(\lambda_{1,2})}{d\lambda} - \frac{dA_1^2(\lambda_{1,2})}{d\lambda}}{\frac{dB_1^2(\lambda_{1,2})}{d\lambda} - D_{opt}^2\frac{dB_2^2(\lambda_{1,2})}{d\lambda}}} \tag{14}$$

Considering the two fixed points, the optimal tuning damping ratio $\zeta_{d\infty}$ is taken as $\zeta_{d\infty} = \sqrt{\frac{\zeta_{d\infty 1}^2 + \zeta_{d\infty 2}^2}{2}}$. Substituting Equations (9) and (12) into Equation (14), $\zeta_{d\infty}$ is analytically obtained, as shown in Equation (15).

$$\zeta_{d\infty} = \sqrt{\frac{\widetilde{\theta}_1^2\theta_2(2\widetilde{\gamma}_1\gamma_4 - \widetilde{\gamma}_2\widetilde{\gamma}_3) - \widetilde{\gamma}_3\theta_0\widetilde{\theta}_1(\widetilde{\gamma}_3\theta_2 + 2\gamma_4\widetilde{\theta}_1) - D_{opt}^2\widetilde{\gamma}_1(2\widetilde{\gamma}_1\gamma_4 - \widetilde{\gamma}_2\widetilde{\gamma}_3)(\widetilde{\gamma}_1\theta_2 - \widetilde{\gamma}_3\theta_0)}{\widetilde{\gamma}_3^2\widetilde{\theta}_1(\widetilde{\theta}_1^2 - \widetilde{\gamma}_1^2 D_{opt}^2)}} \tag{15}$$

The flow chart for determining the analytical optimal parameters via FPA is summarized in Figure 3. Based on the procedure, the analytical solutions to the optimal parameters

{$\nu_\infty$, $\zeta_{d\infty}$} for various IVAs are obtained, as shown in Table 3. As the solution of a general VTMDI is complex, it is given in Appendix A.

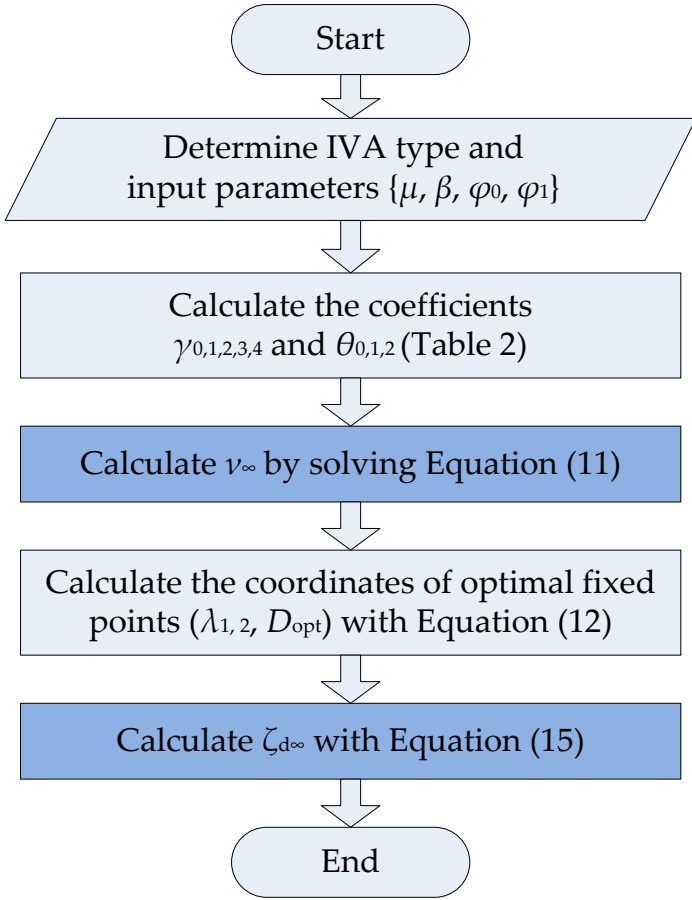

**Figure 3.** A flow chart for determining the analytical optimal parameters via FPA (the procedures that output optimal parameters are highlighted).

**Table 3.** Analytical solutions to the optimal parameters {$\nu_\infty$, $\zeta_{d\infty}$} of various IVAs obtained based on FPA.

| IVA | Parameter | $\nu_\infty$ | $\zeta_{d\infty}$ |
|---|---|---|---|
| TMD [18,19] | $\beta = 0$, $\varphi_0 = 1$, $\varphi_1 = 0$ | $\frac{1}{1+\mu}$ | $\sqrt{\frac{3\mu}{8(1+\mu)}}$ |
| TMDI [28,39] | Arbitrary {$\mu$, $\beta$, $\varphi_0$, $\varphi_1$} | $\frac{\sqrt{1+\frac{\mu}{\mu+\beta}\beta\varphi_1^2}}{1+\mu\varphi_0^2+\beta(\varphi_0-\varphi_1)^2}$ | $\sqrt{\frac{3}{8}\frac{\mu\varphi_0^2+\beta(\varphi_0-\varphi_1)^2-\frac{\mu}{\mu+\beta}\beta\varphi_1^2}{1+\mu\varphi_0^2+\beta(\varphi_0-\varphi_1)^2}}$ |
| TMDI | $\mu\varphi_1 = 0$ (TID [32,33], or grounded TMDI [25–27]) | $\frac{1}{1+\mu_{eq}}$ | $\sqrt{\frac{3\mu_{eq}}{8(1+\mu_{eq})}}$ |
| VTMD [34,35] | $\beta = 0$, $\varphi_0 = 1$, $\varphi_1 = 0$ | $\sqrt{\frac{1}{1-\mu}}$ | $\sqrt{\frac{3\mu}{4(2-\mu)}}$ |
| VTMDI [28,39] | Arbitrary {$\mu$, $\beta$, $\varphi_0$, $\varphi_1$} | Equation (A1) | Equation (A2) |
| VTMDI | $\varphi_1 = 0$ (grounded VTMDI [36]) | $\sqrt{\frac{1}{1-(\mu+\beta)\varphi_0^2}}$ | $\sqrt{\frac{3(\mu+\beta)\varphi_0^2}{4[2-(\mu+\beta)\varphi_0^2]}}$ |
| VTMDI | $\mu = 0$ (TVMD [21], TID2 [37], or VTID [39]) | $\sqrt{\frac{1}{1-\beta(\varphi_0-\varphi_1)^2}}$ | $\sqrt{\frac{3\beta(\varphi_0-\varphi_1)^2}{4[2-\beta(\varphi_0-\varphi_1)^2]}}$ |
| VTMDI | $\mu\varphi_1 = 0$ | $\sqrt{\frac{1}{1-\mu_{eq}}}$ | $\sqrt{\frac{3\mu_{eq}}{4(2-\mu_{eq})}}$ |

It is noted that there are many optimization goals of the DVAs, such as the $H_\infty$ optimization, which aims at minimizing the maximum of the DAF. The $H_2$ optimization

is targeted for minimizing the frequency domain integration, which corresponds to the variance of the response based on the stochastic vibration theory. In the presented paper, we adopted the FPA for an $H_\infty$ optimal solution considering the two reasons. Firstly, the optimal results of the $H_\infty$ and $H_2$ optimization are similar for stationary stochastic vibration responses according to previous studies [29,39]. Secondly, based on the fixed-point approach, the closed form solution can be derived, providing a feasible formula for practical design.

Notice that, in the proposed analytical derivation with FPA, the results for the optimal parameters ($\nu_\infty$ and $\zeta_{d\infty}$) of Equations (11), (12) and (15) are only based on the coefficients of the transfer function. No extra assumption is introduced. Therefore, the derivation can be applied for DVAs that follows a transfer function with a quadratic numerator polynomial and a quartic denominator polynomial. A linear DVA with a single DOF usually follows this characteristic. The abovementioned equations can be applied to such DVAs other than the IVAs investigated in this paper.

It is indicated that when $\mu\varphi_1 = 0$ (i.e., absent mass $\mu = 0$ or grounded IVA $\varphi_1 = 0$), the optimal parameters of the TMDI (or VTMDI) can be formulated with an equivalent mass ratio $\mu_{eq}$ compared to the corresponding conventional TMD (or VTMD). The equivalent mass ratio $\mu_{eq}$ is formulated in Equation (16), revealing the influence of installation locations. This equivalent mass ratio approach can be extended to IVAs for approximating the optimal parameters neglecting the higher order items, as displayed in Equations (17) and (18).

$$\mu_{eq} = \mu\varphi_0^2 + \beta(\varphi_0 - \varphi_1)^2 \tag{16}$$

$$\nu_{opt}^{TMDI} = \frac{1}{1 + \mu_{eq}}; \ \zeta_{dopt}^{TMDI} = \sqrt{\frac{3\mu_{eq}}{8(1 + \mu_{eq})}} \tag{17}$$

$$\nu_{opt}^{VTMDI} = \sqrt{\frac{1}{1 - \mu_{eq}}}; \ \zeta_{dopt}^{VTMDI} = \sqrt{\frac{3\mu_{eq}}{4(2 - \mu_{eq})}} \tag{18}$$

## 3. Wind-Induced Response Estimation

With the determined IVA parameters, the estimation method of the controlled wind-induced response based on the filter approach is presented in this section, which provides a basis to the formulation of ESWL.

### 3.1. Filter-Based Wind Load Spectrum

As an excitation input, the wind load spectrum is important for estimating the wind-induced responses. In order to obtain a closed form solution, a filter-based model is adopted to the estimation of the wind load spectrum. It is described as an analog filter applied on white-noise. The normalized generalized wind load spectrum is written as Equation (19). In the equation, $\Delta(s)$ is a filter polynomial for wind load, and $\delta$ is a normalization factor, determined by $\delta = \left[ \int_0^\infty |\Delta(i\omega)|^{-2} d\omega \right]^{-1}$.

$$\frac{S_F(\omega)}{\sigma_F^2} = \frac{\delta}{|\Delta(i\omega)|^2} \tag{19}$$

For the along-wind excitation characterized by a broad-banded spectrum, the filter polynomial is modeled by a linear function. The filter polynomial and the corresponding normalization factor are given by Equation (20). The subscript "a" stands for "along-wind". In the model, $\omega_a = \text{argmax}\left[\omega S_{Fa}(\omega)/\sigma_{Fa}^2\right]$ is the characteristic frequency of the along-wind load. The model agrees well with the experimental data for tall buildings or slender structures with different configurations in literatures, as shown in Figure 4a.

$$\Delta_a(s) = s/\omega_a + 1; \ \delta_a = \frac{2}{\pi\omega_a} \tag{20}$$

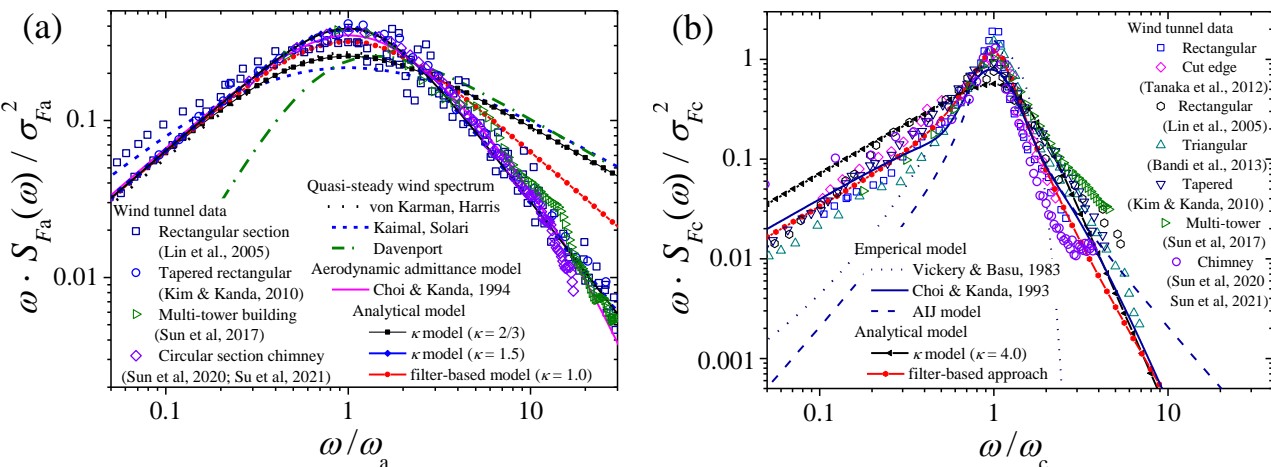

**Figure 4.** Comparison between the spectral model and experimental data reported in relevant literatures [45–55]. (**a**) along-wind; (**b**) cross-wind.

The cross-wind excitation usually has a narrow-banded spectral characteristic, due to the vortex-induced turbulence. A quadratic polynomial is adopted to the model of the cross-wind spectrum, provided by Equation (21), with the subscript "c" denoting a "cross-wind". In the equation, $\omega_c = \mathrm{argmax}\left[\omega S_{Fc}(\omega)/\sigma_{Fc}^2\right]$ is a characteristic frequency of the cross-wind load, related with the vortex frequency. $\lambda_c = \frac{1}{\omega_c}\sqrt{\int_0^\infty \frac{\omega^2 S_F(\omega)}{\sigma_F^2}\mathrm{d}\omega}$ is the bandwidth parameter ranging from 1 to infinity. $\rho_c = \sqrt{\left(\lambda_c^2 + 1\right)^2 - 4}$ is a dissipation parameter. Note that, as the bandwidth becomes increasingly narrow, $\lambda_c$ approaches 1. Otherwise, as $\lambda_c$ tends to infinity, the result converges to the along-wind spectrum. This model agrees well with the experimental data for various slender structures reported in the literatures, as shown in Figure 4b.

$$\Delta_c(s) = (s/\omega_c)^2 + \rho_c s/\omega_c + \lambda_c^2; \ \delta_c = \frac{2}{\pi\omega_c}\rho_c\lambda_c^2 \tag{21}$$

There are many advantages for adopting the filter approach for the wind load spectra [29,42,55]. In addition to the well goodness of fit with experimental data, the major advantage of the filter approach is its simplicity in describing the physical meaning of the spectral characteristics of wind load. Moreover, it also leads to a closed form solution to the wind-induced responses, which significantly enhanced the calculation efficiency and simplicity for the analytical derivation.

### 3.2. Closed Form Solutions to Wind-Induced Responses Based on Filter Approach

With the filter-based wind load spectra, the dynamic wind-induced responses can be calculated through the frequency domain integral, as shown in Equation (3). With the filter approach, the frequency domain integration formatted with Equation (21) can be analytically solved [51]. In this format, the filter polynomial to the $n$th order is provided by $\Lambda(s) = \sum_{j=0}^{n} \chi_j(s/\omega_n)^j$, with $\chi_j$ ($j = 0, 1, 2, \ldots, n$) being the dimensionless filter coefficients. The numerator polynomial of the even degree is less than the $2(n–1)$th order, written as, $\Xi(\omega^2) = \sum_{j=0}^{n-1} \xi_j(\omega/\omega_n)^{2j}$, with $\xi_j$ ($j = 0, 1, 2, \ldots, n–1$) being the dimensionless numerator coefficients.

$$I = \int_0^\infty \frac{\Xi(\omega^2)}{|\Lambda(\mathrm{i}\omega)|^2}\mathrm{d}\omega = \frac{\pi\omega_n}{2\chi_n} \cdot \frac{N}{D} \tag{22}$$

The solution to the integration *I* is expressed as two determinants of the *n*th order, *D* and *N*. They are calculated by Equations (A5) and (A6) (see Appendix B), respectively. *D* is the denominator determinant composed of filter coefficients $\chi_j$. Whereas, *N* is the numerator determinant, similar to *D*, merely replacing the first row with the numerator coefficients $\xi_j$. With this approach, the closed form solutions are obtained in this section.

### 3.2.1. Uncontrolled Response

Substituting uncontrolled $H(s)$ (Equation (2)), and the filter-based wind load spectral model (Equation (19)) into Equation (3), the dimensionless response (with subscript "0" representing uncontrolled) $\beta_{d0} = \frac{\sigma_{x0}}{\sigma_F/K}$ is obtained by Equation (23). The factor $\beta_{d0}$ denotes the ratio between the dynamic response and quasi-static (also known as "background") response.

$$\beta_{d0} = \frac{\sigma_{x0}}{\sigma_F/K} = \sqrt{\delta \cdot \int_0^\infty \frac{1}{|\Delta(i\omega) \cdot V(i\omega)|^2} d\omega} = \sqrt{\kappa_0 \cdot \frac{N_0}{D_0}} \tag{23}$$

For computing the uncontrolled along-wind and cross-wind responses, Equations (20) and (21) are substituted into Equation (23). The filter polynomial $\Lambda(s) = \Delta(s) \cdot V(s)$ is cubic for along-wind response; meanwhile, it is quartic for the cross-wind situation. The numerator polynomial is constant, i.e., $\Xi(\omega^2) \equiv 1$. The dimensionless filter coefficients and the integral factor $\kappa_0 = \frac{\pi \delta \omega_n}{2\chi_n}$ are summarized in Table 4, with $\Omega_{a,c} = \omega_n/\omega_{a,c}$ being the frequency ratio. The determinants $N_0$ and $D_0$ are calculated through Equations (A5) and (A6), as shown in Equation (24). In the equation, coefficients $\psi_1$, $\psi_2$, and $\psi_{12}$ for along-wind and cross-wind (denoted by superscripts "a" and "c") are given by Equations (25) and (26), respectively. Note that, for cross-wind response, the aerodynamic damping ratio $\zeta_a$ should be considered. Consequently, the dimensionless responses are obtained and generically described in Equation (24).

$$\begin{cases} N_0 = \psi_2 \\ D_0 = \psi_1\psi_2 - \psi_{12}^2 \end{cases} \tag{24}$$

$$\begin{cases} \psi_1^a = 1 + 2\zeta_n/\Omega_a \\ \psi_{12}^a = 1 \\ \psi_2^a = 1 + 2\zeta_n\Omega_a \end{cases} \tag{25}$$

$$\begin{cases} \psi_1^c = \frac{\rho_c\lambda_c^2}{\Omega_c}\left[1 + 2(\zeta_n + \zeta_a)\frac{\lambda_c^2}{\rho_c\Omega_c}\right] \\ \psi_{12}^c = 1 + 2(\zeta_n + \zeta_a)\Omega_c/\rho_c \\ \psi_2^c = \frac{\Omega_c}{\rho_c\lambda_c^2}\left\{1 + 2(\zeta_n + \zeta_a)\frac{\Omega_c^2}{\lambda_c^2} \cdot \left[2(\zeta_n + \zeta_a) + \frac{\Omega_c}{\rho_c} + \frac{\rho_c}{\Omega_c}\right]\right\} \end{cases} \tag{26}$$

**Table 4.** Dimensionless filter coefficients for the uncontrolled wind-induced responses.

| Coefficient | Along-Wind | Cross-Wind |
|:---:|:---:|:---:|
| $\chi_0$ | 1 | $\lambda_c^2$ |
| $\chi_1$ | $\Omega_a + 2\zeta_n$ | $\rho_c\Omega_c + 2(\zeta_n + \zeta_a)\lambda_c^2$ |
| $\chi_2$ | $2\zeta_n\Omega_a + 1$ | $\Omega_c^2 + 2(\zeta_n + \zeta_a)\rho_c\Omega_c + \lambda_c^2$ |
| $\chi_3$ | $\Omega_a$ | $2(\zeta_n + \zeta_a)\Omega_c^2 + \rho_c\Omega_c$ |
| $\chi_4$ | — | $\Omega_c^2$ |
| $\xi_0$ | 1 | 1 |
| $\kappa_0$ | 1 | $\rho_c\lambda_c^2/\Omega_c$ |

### 3.2.2. Controlled Response

Similarly, the controlled responses (without the subscript "0") are calculated by Equation (27), substituting the controlled $H(s)$ (Equation (7)) and the filter-based spectra Equation (19) into Equation (22).

$$\beta_{\mathrm{d}} = \frac{\sigma_x}{\sigma_F / K} = \sqrt{\delta \cdot \int_0^\infty \frac{|\Theta(\mathrm{i}\omega)|^2}{|\Delta(\mathrm{i}\omega) \cdot \Gamma(\mathrm{i}\omega)|^2} \mathrm{d}\omega} = \sqrt{\kappa \cdot \frac{N}{D}} \tag{27}$$

The order of the filter polynomial $\Lambda(s) = \Delta(s) \cdot \Gamma(s)$ is $n = 5$ for the along-wind response and $n = 6$ for the cross-wind response. The numerator polynomial in Equation (22) is to the 4th order, i.e., $\Xi(\omega^2) = |\Theta(\mathrm{i}\omega)|^2$. The dimensionless coefficients and integral factor $\kappa = \frac{\pi \delta \omega_{\mathrm{n}}}{2 \chi_n}$ are summarized in Table 5. Comparing with Table 4 for uncontrolled responses, it is indicated that, $\kappa = \gamma_4 \kappa_0$ for controlled responses. The dimensionless responses are obtained with the filter approach in Equation (22). The determinants $N$ and $D$ are calculated from Equation (A5) and (A6). Moreover, the aerodynamic damping ratio $\zeta_{\mathrm{a}}$ should be considered in the cross-wind situation.

**Table 5.** Dimensionless filter coefficients for the controlled wind-induced responses.

| Coefficient | Along-Wind | Cross-Wind |
|:---:|:---:|:---:|
| $\chi_0$ | $\gamma_0$ | $\gamma_0 \lambda_{\mathrm{c}}^2$ |
| $\chi_1$ | $\gamma_0 \Omega_{\mathrm{a}} + \gamma_1$ | $\gamma_0 \rho_{\mathrm{c}} \Omega_{\mathrm{c}} + \gamma_1 \lambda_{\mathrm{c}}^2$ |
| $\chi_2$ | $\gamma_1 \Omega_{\mathrm{a}} + \gamma_2$ | $\gamma_0 \Omega_{\mathrm{c}}^2 + \gamma_1 \rho_{\mathrm{c}} \Omega_{\mathrm{c}} + \gamma_2 \lambda_{\mathrm{c}}^2$ |
| $\chi_3$ | $\gamma_2 \Omega_{\mathrm{a}} + \gamma_3$ | $\gamma_1 \Omega_{\mathrm{c}}^2 + \gamma_2 \rho_{\mathrm{c}} \Omega_{\mathrm{c}} + \gamma_3 \lambda_{\mathrm{c}}^2$ |
| $\chi_4$ | $\gamma_3 \Omega_{\mathrm{a}} + \gamma_4$ | $\gamma_2 \Omega_{\mathrm{c}}^2 + \gamma_3 \rho_{\mathrm{c}} \Omega_{\mathrm{c}} + \gamma_4 \lambda_{\mathrm{c}}^2$ |
| $\chi_5$ | $\gamma_4 \Omega_{\mathrm{a}}$ | $\gamma_3 \Omega_{\mathrm{c}}^2 + \gamma_4 \rho_{\mathrm{c}} \Omega_{\mathrm{c}}$ |
| $\chi_6$ | — | $\gamma_4 \Omega_{\mathrm{c}}^2$ |
| $\tilde{\zeta}_0$ | $\theta_0^2$ | $\theta_0^2$ |
| $\tilde{\zeta}_1$ | $\theta_1^2 - 2\theta_0 \theta_2$ | $\theta_1^2 - 2\theta_0 \theta_2$ |
| $\tilde{\zeta}_2$ | $\theta_2^2$ | $\theta_2^2$ |
| $\kappa$ | $\gamma_4$ | $\gamma_4 \rho_{\mathrm{c}} \lambda_{\mathrm{c}}^2 / \Omega_{\mathrm{c}}$ |

## 4. Equivalent Static Wind Load

The structural design requires ESWL to estimate the peak wind-induced responses, which may be combined with the other load effects. The basic equivalent and generalized wind force $F_{\mathrm{eq}}$ targeting the top displacement of the building is formulated in Equation (28).

$$F_{\mathrm{eq}} = K\hat{x} = K(\overline{x} + g\sigma_x) = \overline{F} + g\beta_{\mathrm{d}}\sigma_F \tag{28}$$

### 4.1. Gust Response Factor for Along-Wind ESWL

For along-wind ESWL, in light of the basic idea of the Davenport's Gust Loading Factor approach [1], adopting the quasi-steady assumption that $\sigma_F / \overline{F} = 2 r I_u$ (with $I_u$ being the turbulence intensity, and $r$ being a modification factor), the gust response factor $G$ may be provided by Equation (29). Furthermore, the equivalent static wind pressure $p_{\mathrm{eq}}(z)$ may be provided by Equation (30), where $\overline{p}(z)$ is the time-averaged wind pressure of $p(z, t)$.

$$G = F_{\mathrm{eq}} / \overline{F} = 1 + g\beta_{\mathrm{d}} \frac{\sigma_F}{\overline{F}} = 1 + 2 r g I_u \beta_{\mathrm{d}} \tag{29}$$

$$p_{\mathrm{eq}}(z) = G \cdot \overline{p}(z) \tag{30}$$

In Equation (29), $\beta_{\mathrm{d}}$ is a factor that considers the dynamic effect. For an uncontrolled structure, it is taken as $\beta_{\mathrm{d}0}$ in Equation (23). While controlled with IVA, it should be calculated by Equation (27). In order to make this process explicit, a control factor is defined as the ratio between controlled $\beta_{\mathrm{d}}$ and uncontrolled $\beta_{\mathrm{d}0}$, as calculated by Equation (31). In

this manner, the relationship between the controlled $G$ and the uncontrolled $G_0$ is provided by Equation (32).

$$J = \frac{\beta_d}{\beta_{d0}} = \sqrt{\frac{\gamma_4 N D_0}{N_0 D}} \qquad (31)$$

$$G = 1 + 2rg I_u \beta_{d0} J = 1 + J(G_0 - 1) \qquad (32)$$

### 4.2. Cross-Wind ESWL

For the cross-wind response, the static response can be neglected. The wind-induced response is dominated by the dynamic component. Moreover, the quasi-steady approach is not applicable. Ignoring the mean component in Equation (28), it is obtained that $F_{eq} = g\beta_d \sigma_F$. There are two approaches to estimate the ESWL.

The first one is the extended gust response factor $G'$, as referred to in [6], in which the mean along-wind load $\overline{F}_a$ is used, as shown in Equation (33). The subscript "a" and "c" represent "along-wind" and "cross-wind", respectively. Consequently, the equivalent static wind pressure in the cross-wind direction, $p_{eq,c}(z)$, may be given by Equation (34) in proportion to the mean along-wind load $\overline{p}_a(z)$.

$$G' = \frac{F_{eq,c}}{\overline{F}_a} = g\beta_d \frac{\sigma_{Fc}}{\overline{F}_a} \qquad (33)$$

$$p_{eq,c}(z) = G' \cdot \overline{p}_a(z) \qquad (34)$$

Alternatively, another approach directly adopts the inertial load of the fundamental mode, which is expressed by Equation (35). The equivalent static wind pressure in the cross-wind direction, $p_{eq,c}(z)$, is in proportion to the modal function $\Phi(z)$.

$$p_{eq,c}(z) = gK\sigma_x \Phi(z) = g\beta_d \sigma_{Fc} \Phi(z) \qquad (35)$$

In either method, the cross-wind ESWL is in proportion to $\beta_d$. The relationship between the controlled load $p_{eq,c}$ and the uncontrolled load $p_{eq,c0}$ is provided by Equation (36). The control ratio $J$ is estimated from Equation (32) using the cross-wind spectrum.

$$p_{eq,c} = J \cdot p_{eq,c0} \qquad (36)$$

It is noticed that, with the proposed framework, the ESWL of the uncontrolled structure can be converted to the controlled ESWL using the control ratio $J$. The ratio $J$ is dependent on the spectral parameters of the wind load and the tuning parameters of the IVA. Note that the control ratios for along-wind and cross-wind responses should be calculated separately. With this approach, it is also convenient to convert the codified ESWL for uncontrolled structures to that for the controlled structures.

## 5. Case Study

In this section, a numerical case study on the wind-induced vibration control and ESWL estimation of a tall chimney with IVA is performed to demonstrate the application of the proposed procedure.

The chimney is made of reinforced concrete, with a height of $H = 270$ m. It is assumed to be constructed in an open terrain (Type C, ASCE). The sectional dimensions along the height are listed in Table 6. The averaged outer diameter of the chimney is 25.4 m. The finite element of the chimney is established with a beam-type element. Through a modal analysis, the fundamental frequency of the chimney is calculated as $\omega_n = 2.48$ rad/s. The corresponding modal function $\Phi(z)$ is also presented in Table 6. The modal mass is $M = 4588$ t. The critical wind velocity of the chimney is determined as $U_{Cr} = 50.2$ m/s. The critical damping ratio of the chimney is $\zeta_n = 1.5\%$. According to the wind tunnel tests on an aeroelastic model [43], the aerodynamic damping ratio at $U_{Cr}$ is $\zeta_a = -0.96\%$. In the

numerical case, the most unfavorable case is considered, as the design wind velocity is equal to the critical wind velocity.

**Table 6.** Geometric profile and modal function of the chimney.

| Height (m) | Outer Diameter (m) | Thickness (m) | Modal Function $\Phi(z)$ |
|---|---|---|---|
| 270 | 16.9 | 0.45 | 1.000 |
| 260 | 17.3 | 0.45 | 0.933 |
| 250 | 17.7 | 0.45 | 0.866 |
| 240 | 18.1 | 0.50 | 0.799 |
| 230 | 18.5 | 0.50 | 0.733 |
| 220 | 18.9 | 0.50 | 0.668 |
| 210 | 19.3 | 0.50 | 0.605 |
| 200 | 19.7 | 0.55 | 0.543 |
| 180 | 20.5 | 0.55 | 0.429 |
| 160 | 22.1 | 0.60 | 0.328 |
| 140 | 23.7 | 0.65 | 0.240 |
| 120 | 25.3 | 0.65 | 0.168 |
| 100 | 26.9 | 0.70 | 0.110 |
| 80 | 28.9 | 0.80 | 0.066 |
| 40 | 33.7 | 0.90 | 0.015 |
| 0 | 38.5 | 1.00 | 0.000 |

For the IVA, it is assumed that the mass ratio is $\mu = 1.0\%$, and the inertance ratio is $\beta = 20\%$. The IVA is installed between $z_0 = 260$ m ($\varphi_0 = 0.933$) and $z_1 = 210$ m ($\varphi_1 = 0.605$). Using Equations (16)–(18), the optimal parameters of the IVAs are calculated, as shown in Table 7. Here, the TMDI and VTMDI cases are considered. The modulus of the frequency response functions $|X(i\omega)/F(i\omega)|$ of uncontrolled and controlled chimney cases are shown in Figure 5. The theoretical curves (using the optimal parameters in Table 3) and the proposed ones (obtained with parameters in Table 7, obtained by Equations (17) and (18)) are compared in the figures. The results have demonstrated good agreements between the proposed formulas and the theoretical curves on the chimney case, indicating the effectiveness of the proposed optimal design method.

**Table 7.** Determined IVA parameters for the chimney case.

| Parameter | TMDI | VTMDI |
|---|---|---|
| $\mu_{eq}$ (%) | 3.15 | 3.15 |
| $\nu_{opt}$ | 0.969 | 1.016 |
| $\zeta_{dopt}$ (%) | 10.71 | 10.96 |
| $m$ (ton) | 45.88 | 45.88 |
| $c$ (kN·s/m) | 496 | 532 |
| $k$ (kN/m) | 5569 | 6119 |
| $b$ (kN·s$^2$/m) | 917.6 | 917.6 |

Based on the wind tunnel data [47,48], the wind-induced responses of the uncontrolled and controlled cases are calculated with time-history analysis. The time-history of the wind-induced top displacement responses are shown in Figure 6. The statistical results of the peak wind-induced responses are summarized in Table 8. The resulting control ratios and gust response factors obtained from the proposed ESWL framework are also shown in this table. A good agreement can be observed between the time domain method and the proposed method. The proposed method may overestimate the GRFs within a maximum relative error of 3.9% in the numerical cases, which is an acceptable error in practical engineering. Therefore, the effectiveness of the analytical framework regarding the optimal design and control performance estimation for the ESWL of structures with IVAs in this paper are illustrated.

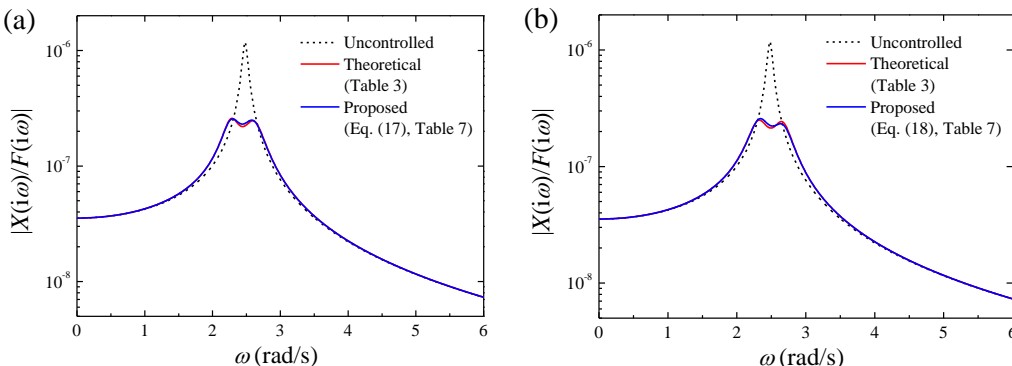

**Figure 5.** The modulus of frequency response functions $|X(i\omega)/F(i\omega)|$ of uncontrolled and controlled chimney cases. (**a**) TMDI. (**b**) VTMDI.

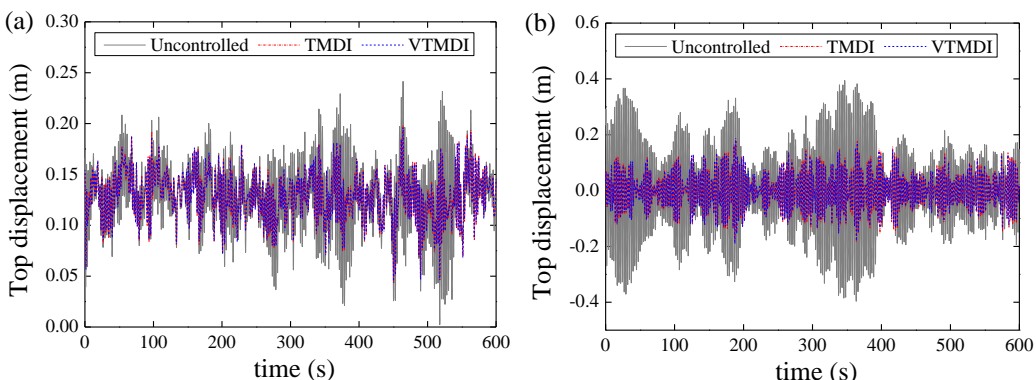

**Figure 6.** The time-history of wind-induced top displacement responses of uncontrolled and controlled chimney cases. (**a**) Along-wind. (**b**) Cross-wind.

**Table 8.** Peak wind-induced responses, control ratios, and gust response factors for the cases.

| Direction | Parameter | Uncontrolled | TMDI | VTMDI |
|---|---|---|---|---|
| | Top displacement (m) | 0.245 | 0.207 | 0.206 |
| | Base shear force (MN) | 17.72 | 15.50 | 15.46 |
| | Base bending moment (GN·m) | 2.793 | 2.388 | 2.381 |
| Along-wind | Control ratio $J$ (time domain method) | — | 0.676 | 0.670 |
| | Gust Response Factor $G$ (time domain method) | 1.93 | 1.63 | 1.62 |
| | Control ratio $J$ (proposed method) | — | 0.692 | 0.680 |
| | Gust Response Factor $G$ (proposed method) | 1.95 | 1.66 | 1.64 |
| | Top displacement (m) | 0.422 | 0.197 | 0.194 |
| | Base shear force (MN) | 19.36 | 9.14 | 8.95 |
| | Base bending moment (GN·m) | 4.033 | 1.901 | 1.868 |
| Cross-wind | Control ratio $J$ (time domain method) | — | 0.467 | 0.461 |
| | Gust Response Factor $G'$ (time domain method) | 3.32 | 1.55 | 1.53 |
| | Control ratio $J$ (proposed method) | — | 0.476 | 0.469 |
| | Gust Response Factor $G'$ (proposed method) | 3.38 | 1.61 | 1.59 |

Moreover, it can be also concluded from the wind-induced vibration control analysis results that the control ratio for VTMDI is slightly lower than that for TMDI, indicating a better control performance. However, it requires a larger stiffness and damping to maintain an optimal design state. Therefore, the designers can make a choice according to the conclusions and methods proposed in this paper. Moreover, the control ratios for cross-wind responses are lower than those for along-wind responses. This is because the cross-wind response is subjected to a more significant dynamic effect due to the vortex-induced resonance. Generally, according to the time domain analysis, with the IVA, the ESWL denoted

by the gust response factors for the along-wind was reduced by approximately 15%. It reduced up to 53% for the cross-wind ESWL. The IVAs appear to be more effective to suppress the vortex-induced resonant vibration of the structures with low damping. When controlled by IVAs, the ESWL of vortex resonance become less significant, which is reduced to the along-wind ESWL, leading to an economic design for practical engineering structures.

## 6. Conclusions

In this paper, an analytical framework of the Equivalent Static Wind Load (ESWL) for structures with Inerter-based Vibration Absorbers (IVAs) was established. This framework includes analytical parametric optimization formulas based on the Fixed-point approach (FPA), closed form solutions for the controlled wind-induced responses based on the filter approach, and ESWL for the controlled structures based on the gust response factors.

The core of the proposed ESWL for controlled structures is based on a control ratio. It is dependent on the spectral parameters of wind load and the tuning parameters of the IVAs. A closed form solution of the control ratio is presented. With the control ratio, the original uncontrolled ESWL can be easily converted to the controlled one, which will provide a quick estimation at the preliminary design stage.

The current investigation is applicable to structures dynamically dominated by a single mode. It can be extended to more complex structures in future studies. Moreover, the presented analytical approach for IVAs is based on a general mathematical model of a transfer function with a quartic filter polynomial; it can also be applied to other DVAs with similar features, such as negative-stiffness-based DVAs, and so on.

**Author Contributions:** Conceptualization, N.S.; methodology, Z.C.; software, N.H.; validation, N.S., N.H. and Z.C.; formal analysis, N.S.; investigation, N.S.; resources, S.P.; data curation, Z.C.; writing—original draft preparation, N.S.; writing—review and editing, Y.U.; visualization, N.S.; supervision, Y.U.; project administration, S.P.; funding acquisition, S.P. All authors have read and agreed to the published version of the manuscript.

**Funding:** This research was funded by the National Key R&D Program of China (Grant No. 2022YFB2602302), National Natural Science Foundation of China (Grant Nos. 52202415, 51878129), Tianjin Science and Technology Development Project (Grant No. 22YFZCSN00030), and Tianjin Transportation Science and Technology Development Plan Project (Grant No. 2022-23).

**Institutional Review Board Statement:** Not applicable.

**Informed Consent Statement:** Not applicable.

**Data Availability Statement:** Not applicable.

**Conflicts of Interest:** The authors declare no conflict of interest.

## Appendix A. Analytical Solutions of $\{\nu_\infty, \zeta_{d\infty}\}$ for VTMDI

In Table 3, the analytical solutions to the optimal parameters $\{\nu_\infty, \zeta_{d\infty}\}$ of VTMDI based on FPA are shown in Equations (A1) and (A2), where the polynomials $\Psi_1$ and $\Psi_2$ are shown in Equations (A3) and (A4).

$$\nu_\infty = \sqrt{\frac{1 + \frac{\mu}{\mu+\beta}\beta\varphi_1^2}{1 + \mu\varphi_0\varphi_1(3 + \mu\varphi_0^2) - \left[\mu\varphi_0^2 + \beta(1 + \mu\varphi_1^2)(\varphi_0 - \varphi_1)^2\right]}} \tag{A1}$$

$$\zeta_{d\infty} = 2\nu_\infty(1 + \mu\varphi_1^2)\sqrt{\frac{(\mu + \beta)\left[2 + \nu_\infty^2(\mu + \beta)(\varphi_0 - \varphi_1)^2\right]}{\nu_\infty^4(\mu + \beta)(\varphi_0 - \varphi_1)^2\Psi_1 + \nu_\infty^2\Psi_2 + 6(\mu + \beta + \mu\beta\varphi_1^2)}} \tag{A2}$$

$$\begin{aligned}\Psi_1 = {}&5\beta^2(\varphi_0 - \varphi_1)^2(1 + \varphi_1^2\mu) + \mu^3\varphi_0\varphi_1^2(\varphi_0 - 2\varphi_1) + 2\mu^2\beta\varphi_1^2(3\varphi_0^2 - 5\varphi_0\varphi_1 + \varphi_1^2) + \\ &\mu^2\varphi_0(5\varphi_0 - 8\varphi_1) + 2\mu\beta(5\varphi_0^2 - 9\varphi_0\varphi1 + 2\varphi_1^2) - 2(\mu + \beta)\end{aligned} \tag{A3}$$

$$\Psi_2 = 11\beta^2(\varphi_0 - \varphi_1)^2(1 + \varphi_1^2\mu) + \mu^3\varphi_0\varphi_1^2(\varphi_0 - 4\varphi_1) + \mu\varphi_0(\varphi_0 - 2\varphi_1)\left[11(\mu + 2\beta) + 12\mu\beta\varphi_1^2\right] + 2\mu\varphi_1^2(\mu + 5\beta + 3\mu\beta\varphi_1^2) - 6(\mu + \beta) \tag{A4}$$

### Appendix B. The Determinants of $D$ and $N$ for Filter Approach

In Equation (22), the determinants of $D$ and $N$ are shown in Equations (A5) and (A6), respectively.

$$D = \begin{vmatrix} \chi_{n-1} & -\chi_{n-3} & \chi_{n-5} & -\chi_{n-7} & \cdots & \cdot & \cdot & \cdot \\ -\chi_n & \chi_{n-2} & -\chi_{n-4} & \chi_{n-6} & \cdots & \cdot & \cdot & \cdot \\ \cdot & -\chi_{n-1} & \chi_{n-3} & -\chi_{n-5} & \chi_{n-7} & \cdots & \cdot & \cdot \\ \cdot & \chi_n & -\chi_{n-2} & \chi_{n-4} & -\chi_{n-6} & \cdots & \cdot & \cdot \\ \cdot & \cdot & \chi_{n-1} & -\chi_{n-3} & \chi_{n-5} & -\chi_{n-7} & \cdots & \cdot \\ \cdot & \cdot & \cdot & & & \ddots & & \cdot \\ \cdot & \cdot & \cdot & & & \cdots & \chi_1 & \cdot \\ \cdot & \cdot & \cdot & & & \cdots & -\chi_2 & \chi_0 \end{vmatrix} \tag{A5}$$

$$N = \begin{vmatrix} \xi_{n-1} & \xi_{n-2} & \cdots & & & & \cdots & \xi_0 \\ -\chi_n & \chi_{n-2} & -\chi_{n-4} & \chi_{n-6} & \cdots & \cdot & \cdot & \cdot \\ \cdot & -\chi_{n-1} & \chi_{n-3} & -\chi_{n-5} & \chi_{n-7} & \cdots & \cdot & \cdot \\ \cdot & \chi_n & -\chi_{n-2} & \chi_{n-4} & -\chi_{n-6} & \cdots & \cdot & \cdot \\ \cdot & \cdot & \chi_{n-1} & -\chi_{n-3} & \chi_{n-5} & -\chi_{n-7} & \cdots & \cdot \\ \cdot & \cdot & \cdot & & & \ddots & & \cdot \\ \cdot & \cdot & \cdot & & & \cdots & \chi_1 & \cdot \\ \cdot & \cdot & \cdot & & & \cdots & -\chi_2 & \chi_0 \end{vmatrix} \tag{A6}$$

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
