# Peer review of "Equivalent Static Wind Load for Structures with Inerter-Based Vibration Absorbers"

_2674-032X, doi:10.3390/wind2040040_

Round 1

Reviewer 1 Report

<Major comments>

This paper presents a establishing the Equivalent Static Wind Load (ESWL) for structures with the Inerter-based Vibration Absorber (IVA). This paper explains the theory carefully. However, the examination of the proposed method is insufficient. So, I cannot confirm the effectiveness of the proposed method.

I have some major-revision requests for this paper as shown below;

(1)     L97-98: “the displacement response time-history u(z, t) can be decoupled as a spatial variant normalized modal function Φ(z) with respect to the generalized coordinate (height) z (0 ≤ z ≤ H, Φ(H) = 1)

This Φ is the mode shape with no damping (proportional damping system). However, the target model considered in this paper has the local damper, so the target model is a non-proportional damping system.

Thus, you need to explain the reason (or assuming) that the proportional damping mode shape Φ can use for the non-proportional damping systems in this study.

(2)     L115-116: “The peak factor g is approximated with・・・・ where T is・・・

There is no verification of the peak factor. 

The formula can use for the models with low damping, and this paper focuses on models with high damping. So, I can't confirm if the formula used in this paper is applicable.

You are necessary to show the peak factor obtained from the time history analysis using the model, and indicate whether the value matches this expression.

(3)     Fig.2: Figure 2 shows the theoretical DAF curve. However, it is not shown whether the target model used in this paper matches this theoretical curve, and the verification is insufficient.

You should carry out the time history analysis on the target model, and show the results of the agreement with this theoretical curve.

(4)     Fig. 4: Figure 4 shows a comparison between the proposed formula in this paper and the results of wind tunnel experiments, and the results of previous studies are also presented.

However, there is no explanation that the proposed formula is superior to previous studies.

Therefore, I cannot judge the effectiveness of the proposed formula.

In Figure 4, you should explain the advantages (effectiveness) of the proposed formula over previous studies.

(5)     L329-330: “the wind-induced responses of the uncontrolled and controlled cases are calculated.”

Since the details of the calculation method are not given, I can’t judge the results.

Are the responses obtained from time history response analysis? or spectrum modal analysis?

You should indicate not only the calculation method but also the information of the model (2DOF?) used for time history analysis

(6)     L330-331: “The resulting control ratios and gust response factors obtained from the proposed ESWL framework are also shown in this table.”

Are these results shown in Table 8 calculated by using ESWL proposed in this paper?

Are these results compared with time history analysis results?

Since the calculations step (detail) and parameters are not shown, it is not possible to judge the adequacy of the values shown in the Table 8.

You need to show the progress of the calculations and the values used so that I (or a reader) can reproduce the values in the Table 8, and you should show the accuracy of these results (Table 8) by comparing time history analysis results.

(7)     L336-337: “It is concluded form (from?) the analysis results that the control ratio for VTMDI is slightly lower than that for TMDI, indicating a better control performance.

You need to prove that this setting (Table 7) is the optimal setting for VTMDI and TMDI model This paper presents the biaxial effect on inelastic wind-induced response.

(8)     L164: “The IVAs appears to be more effective to suppress the vortex-induced resonant vibration of structures with low damping. When controlled by IVAs, the ESWL of vortex resonance become less significant, which is reduced to the along-wind ESWL, leading to an economic design for practical engineering structures.

It is a natural conclusion that the response is reduced by installing a damper.

The important thing is to show that the response is as expected (design target) even if this paper is "an analytical framework". However, the design target (Control ratio J?) is not shown in this paper. Therefore, I cannot confirm the effectiveness of the proposed method.

<Minor comment>

(1)     L79: A representative study should be cited for the filter approach

END

Author Response

Response to Reviewer 1

This paper presents a establishing the Equivalent Static Wind Load (ESWL) for structures with the Inerter-based Vibration Absorber (IVA). This paper explains the theory carefully. However, the examination of the proposed method is insufficient. So, I cannot confirm the effectiveness of the proposed method.

 I have some major-revision requests for this paper as shown below;

Reply: Thank you for your comment. We have significantly enhanced the effectiveness analysis in the revised manuscript according to your comments. In the revised version, we provide many supportive results in the case study section. We also provide the requested explanations in the rest sections. The revised parts are marked blue in the uploaded manuscript.

(1)     L97-98: “the displacement response time-history u(z, t) can be decoupled as a spatial variant normalized modal function Φ(z) with respect to the generalized coordinate (height) z (0 ≤ z ≤ H, Φ(H) = 1)

This Φ is the mode shape with no damping (proportional damping system). However, the target model considered in this paper has the local damper, so the target model is a non-proportional damping system.

Thus, you need to explain the reason (or assuming) that the proportional damping mode shape Φ can use for the non-proportional damping systems in this study.

Reply: Thank you for your comment. In order to consider the installation location of IVA in the analytical derivation, the SDOF assumption for the primary structure is adopted. Thus, the displacement response is decoupled as u(z, t) = Φ(z) x(t). Such assumption can be extended for a non-proportional damping structure systems (structure controlled with DVA) for a simplified derivation, which is also called a Ritz-Galerkin method. It is classically applied in linear continuum dynamics problems. Several researches regarding this field (for non- proportional structural damping system with DVAs) adopt this method, such as the following literatures.

  • Wang, A. Giaralis. Enhanced motion control performance of the tuned mass damper inerter through primary structure shaping, Structural Control and Health Monitoring 28(8) (2021) e2756.
  • Su, Y. Xia, S. T. Peng. Filter-based inerter location dependence analysis approach of Tuned mass damper inerter (TMDI) and optimal design. Engineering Structures 250 (2022) 113459.
  • Su, J. Bian, S. T. Peng, Y. Xia. Generic optimal design approach for inerter-based tuned mass systems. International Journal of Mechanical Sciences 233 (2022) 107654.
  • Su, J. Bian, S. T. Peng, Y. Xia. Impulsive resistant optimization design of tuned viscous mass damper (TVMD) based on stability maximization. International Journal of Mechanical Sciences 239 (2023) 107876.

These aspects are supplied in the revised manuscript.

  • (Page 4, Lines 129-130) “For a primary structure controlled by an IVA installed between coordinates z0 and z1, as shown in Fig. 1, the Ritz-Galerkin method is adopted as referred to [29, 39, 43, 44] assuming that u(z, t) = Φ(z)·x(t). According to the principle of visual work, the equations of motion are rewritten as Eq. (4).

(2)     L115-116: “The peak factor g is approximated with・・・・ where T is・・・

There is no verification of the peak factor. 

The formula can use for the models with low damping, and this paper focuses on models with high damping. So, I can't confirm if the formula used in this paper is applicable.

You are necessary to show the peak factor obtained from the time history analysis using the model, and indicate whether the value matches this expression.

Reply: Thank you for your comment. The peak factor approach is a statistical method adopted in wind engineering according to Davenport. This approach classically assumes that the parent distribution as a Gaussian distribution, and the mathematical expectation of the extreme value is derived from the extreme value distribution approach. This theory is widely applied to approximate Gaussian stochastic processes such as wind velocity, wind load, and wind-induced response. In fact, this approach is not related whether the response is with DVAs.

The authors understand that the review concerns that whether the effectiveness of the peak factor can be applied for wind-excited structures with DVAs. In order to address this aspect, in the case study section, we provided the time-history results in section 5.

In order to address the above aspects,

  • (Page 3, Line 117) The literature regarding the peak factor approach is mentioned, “According to Davenport’s statistical approach [1], the peak factor g is …”.
  • (Page 15, Figure 6) The time-history results of the uncontrolled and controlled wind-induced responses of the numerical cases are provided.

Figure 6. The time-history of wind-induced top displacement responses of uncontrolled and controlled chimney cases. (a) along-wind. (b) cross-wind.

(3)     Fig.2: Figure 2 shows the theoretical DAF curve. However, it is not shown whether the target model used in this paper matches this theoretical curve, and the verification is insufficient.

You should carry out the time history analysis on the target model, and show the results of the agreement with this theoretical curve.

Reply: Thank you for your comment. Fig. 2 shows the DAF curve with different DVA parameters ν and ζd. It is the analytical solution. The target of this figure is to demonstrate that when ν = ν, the vertical values of the two fixed points are equal. And, when ζd = ζd∞, the fixed points are assigned to the peaks of the DAF curve. The figure is used to prove that the analytical derivation of the H optimal solutions follows the Den Hartog’s fixed point theory.

The authors understand that review concerns that whether the theory is applicable for the target model. The effectiveness of Den Hartog’s fixed point theory is also mentioned by many related literatures. In order to address this concern, the DAF curve for the numerical case is presented, and the time-history results are given in the case study section.

In order to address the above aspects,

  • (Page 14, Figure 5) A figure is added to compare the frequency response functions for the numerical cases to illustrate the effectiveness of the proposed approach on optimal design.

Figure 5. The modulus of frequency response functions |X(iω)/F(iω)| of uncontrolled and controlled chimney cases. (a) TMDI. (b) VTMDI.

  • (Page 14, Lines 360-366) Several sentences are added to explain the results, “The modulus of frequency response functions |X(iω)/F(iω)| of uncontrolled and controlled chimney cases are shown in Fig. 5. The theoretical curves (using the optimal parameters in Table 3) and the proposed ones (obtained with parameters in Table 7, obtained by Eqs. (17) and (18)) are compared in the figures. The results have shown good agreements between the proposed formulas and the theoretical curves on the chimney case, indicating the effectiveness of the proposed optimal design method.”.

(4)     Fig. 4: Figure 4 shows a comparison between the proposed formula in this paper and the results of wind tunnel experiments, and the results of previous studies are also presented.

However, there is no explanation that the proposed formula is superior to previous studies.

Therefore, I cannot judge the effectiveness of the proposed formula.

In Figure 4, you should explain the advantages (effectiveness) of the proposed formula over previous studies.

Reply: Thank you for your comment. We agreed that the advantage of this approach is not fully addressed in this paper. In fig. 4, it is observed that the adopted filter approach on the spectral model agree well with the experimental data. In addition to the well goodness of fit, the major advantage of the filter approach is its simplicity in describing the physical meaning of spectral characteristics of wind load. Moreover, it also leads to closed form solution to the wind-induced responses, which significantly enhanced the calculation efficiency and simplicity for analytical derivation. These advantages are also mentioned in literatures relevant to the filter approach.

In order to address this aspect, a paragraph is added.

  • (Page 9, Lines 250-255) “There are many advantages for adopting the filter approach for the wind load spectra [29, 42, 55]. In addition to the well goodness of fit with experimental data, the major advantage of the filter approach is its simplicity in describing the physical meaning of spectral characteristics of wind load. Moreover, it also leads to closed form solution to the wind-induced responses, which significantly enhanced the calculation efficiency and simplicity for analytical derivation.”.

(5)     L329-330: “the wind-induced responses of the uncontrolled and controlled cases are calculated.”

Since the details of the calculation method are not given, I can’t judge the results.

Are the responses obtained from time history response analysis? or spectrum modal analysis?

You should indicate not only the calculation method but also the information of the model (2DOF?) used for time history analysis

Reply: Thank you for your comment. The calculation of wind-induced responses for the numerical case is through time-history analysis based on the experimental data referred to [47, 48]. The information on the analysis results are provided in the revised manuscript. The information of the Finite element model and the IVAs are displayed in Tables 6 and 7.

  • (Pages 14-15, Lines 372-382) A paragraph is rewritten to describe the time-domain analysis, “… with time-history analysis. The time-history of wind-induced top displacement responses are shown in Fig. 6. The statistical results of the peak wind-induced responses are summarized in Table 8. The resulting control ratios and gust response factors obtained from the proposed ESWL framework are also shown in this table. A good agreement can be observed between the time domain method and the proposed method. The proposed method may overestimate the GRFs within a maximum relative error of 3.9% in the numerical cases, which is an acceptable error in practical engineering. Therefore, the effectiveness of the analytical framework regarding the optimal design and control performance estimation for the ESWL of structures with IVAs in this paper are illustrated.
  • (Page 15, Figure 6) The time-history of the top displacement results are provided.

Figure 6. The time-history of wind-induced top displacement responses of uncontrolled and controlled chimney cases. (a) along-wind. (b) cross-wind.

(6)     L330-331: “The resulting control ratios and gust response factors obtained from the proposed ESWL framework are also shown in this table.”

Are these results shown in Table 8 calculated by using ESWL proposed in this paper?

Are these results compared with time history analysis results?

Since the calculations step (detail) and parameters are not shown, it is not possible to judge the adequacy of the values shown in the Table 8.

You need to show the progress of the calculations and the values used so that I (or a reader) can reproduce the values in the Table 8, and you should show the accuracy of these results (Table 8) by comparing time history analysis results.

Reply: Thank you for your comment. The results in the original manuscript is based on time domain analysis, the ESWL is calculated with the proposed framework based on the time-domain results. In order to further illustrate the effectiveness of the estimation of control ratio, which is a part of the proposed analytical framework with filter approach through frequency domain, the corresponding results are added in the revised manuscript.

  • (Page 15, Table 8) The control ratio and GRF results based on the proposed analytical framework with filter approach through frequency domain are added for comparison.

Table 8. Peak wind-induced responses, control ratios, and gust response factors for the cases.

Direction

Parameter

Uncontrolled

TMDI

VTMDI

Along-wind

Top displacement (m)

0.245

0.207

0.206

Base shear force (MN)

17.72

15.50

15.46

Base bending moment (GN·m)

2.793

2.388

2.381

Control ratio J (time domain method)

0.676

0.670

Gust Response Factor G (time domain method)

1.93

1.63

1.62

Control ratio J (proposed method)

0.692

0.680

Gust Response Factor G (proposed method)

1.95

1.66

1.64

Cross-wind

Top displacement (m)

0.422

0.197

0.194

Base shear force (MN)

19.36

9.14

8.95

Base bending moment (GN·m)

4.033

1.901

1.868

Control ratio J (time domain method)

0.467

0.461

Gust Response Factor G’ (time domain method)

3.32

1.55

1.53

Control ratio J (proposed method)

0.476

0.469

Gust Response Factor G’ (proposed method)

3.38

1.61

1.59

  • (Pages 14-15, Lines 374-382) A paragraph is added to describe the results and discuss the effectiveness, “The statistical results of the peak wind-induced responses are summarized in Table 8. The resulting control ratios and gust response factors obtained from the proposed ESWL framework are also shown in this table. A good agreement can be observed between the time domain method and the proposed method. The proposed method may overestimate the GRFs within a maximum relative error of 3.9% in the numerical cases, which is an acceptable error in practical engineering. Therefore, the effectiveness of the analytical framework regarding the optimal design and control performance estimation for the ESWL of structures with IVAs in this paper are illustrated.”.

(7)     L336-337: “It is concluded form (from?) the analysis results that the control ratio for VTMDI is slightly lower than that for TMDI, indicating a better control performance.

You need to prove that this setting (Table 7) is the optimal setting for VTMDI and TMDI model This paper presents the biaxial effect on inelastic wind-induced response.

Reply: Thank you for your comment. In this paper, we assume the structure under wind load behave in a linear elastic range, the nonlinear inelastic effect is beyond the scope of this paper. In addition, we assume that the IVAs can be ideally implemented for the designed parameters. The deficiencies and nonlinearities are also important aspects of structural control, which will be included in future investigations.

(8)     L164: “The IVAs appears to be more effective to suppress the vortex-induced resonant vibration of structures with low damping. When controlled by IVAs, the ESWL of vortex resonance become less significant, which is reduced to the along-wind ESWL, leading to an economic design for practical engineering structures.

It is a natural conclusion that the response is reduced by installing a damper.

The important thing is to show that the response is as expected (design target) even if this paper is "an analytical framework". However, the design target (Control ratio J?) is not shown in this paper. Therefore, I cannot confirm the effectiveness of the proposed method.

Reply: Thank you for your comment. We agree that the response reduction is a natural conclusion. Some other studies also support the conclusion that the control performance of VTMDI is better than TMDI. And the IVAs behave better for the lower damping cases.

The authors understand the reviewer’s concern on the validation of the optimal design and performance evaluation.

In this paper, the optimal design is achieved by Den Hartog’s fixed point approach by equalizing the peaks of the frequency response function. For the numerical case, a figure regarding this aspect is added to prove that the proposed method leads to such a state.

  • (Page 14, Figure 5) (Page 14, Figure 5) A figure is added to compare the frequency response functions for the numerical cases to illustrate the effectiveness of the proposed approach on optimal design.

Figure 5. The modulus of frequency response functions |X(iω)/F(iω)| of uncontrolled and controlled chimney cases. (a) TMDI. (b) VTMDI.

  • (Page 14, Lines 360-366) Several sentences are added to explain the results, “The modulus of frequency response functions |X(iω)/F(iω)| of uncontrolled and controlled chimney cases are shown in Fig. 5. The theoretical curves (using the optimal parameters in Table 3) and the proposed ones (obtained with parameters in Table 7, obtained by Eqs. (17) and (18)) are compared in the figures. The results have shown good agreements between the proposed formulas and the theoretical curves on the chimney case, indicating the effectiveness of the proposed optimal design method.”.

In addition, regarding the control performance estimation, the effectiveness validation is provided in the revised manuscript.

  • (Pages 14-15, Lines 374-382) A paragraph is added to describe the results and discuss the effectiveness, “The statistical results of the peak wind-induced responses are summarized in Table 8. The resulting control ratios and gust response factors obtained from the proposed ESWL framework are also shown in this table. A good agreement can be observed between the time domain method and the proposed method. The proposed method may overestimate the GRFs within a maximum relative error of 3.9% in the numerical cases, which is an acceptable error in practical engineering. Therefore, the effectiveness of the analytical framework regarding the optimal design and control performance estimation for the ESWL of structures with IVAs in this paper are illustrated.”.

<Minor comment>

(1)     L79: A representative study should be cited for the filter approach

Reply: Thank you for your comment. We have supplied the citation in Page 2, Line 82.

Reviewer 2 Report

This study investigates the equivalent static wind load for structures with inerter-based

vibration absorbers. The topic is within the scope of this journal. The paper is well organized and written. The paper can be accepted after some minor changes.

1. It is suggested to explain why the four configurations are selected in Figure 1. Indeed, there are several studies comparing the performances of various configurations. The cons and pros of several configurations are already known.

2. It is suggested to discuss how two terminals of an inerter can be separated for a real structure.

3. It is suggested to enrich the introduction with some recent studies on TMD and inerter-based damper, e.g., https://doi.org/10.1016/j.jweia.2021.104836, https://doi.org/10.1016/j.engstruct.2022.115121

Author Response

Response to Reviewer 2

This study investigates the equivalent static wind load for structures with inerter-based vibration absorbers. The topic is within the scope of this journal. The paper is well organized and written. The paper can be accepted after some minor changes.

  1. It is suggested to explain why the four configurations are selected in Figure 1. Indeed, there are several studies comparing the performances of various configurations. The cons and pros of several configurations are already known.

Reply: Thank you for your comment. It is noted that the present paper focus on two major configurations of single-tuned inerter-based vibration absorbers, TMDI and VTMDI, as displayed in Fig. 1. For the four configurations, TMD and TID can be expressed as TMDI with an absent of mass. Likewise, VTMD and TVMD can be VTMDI with an absent of mass. Therefore, these configurations are the major IVAs with similar components and different configurations. They can be formulated and modeled generically. Although the cons and pros are discussed in several papers including the author’s previous study. The ESWL of structures with these IVAs are not fully addressed.

In order to address this aspect, a paragraph is added.

  • (Page 4, Lines 121-127) “In this paper, two major configurations of IVAs, TMDI and VTMDI are considered, as shown in Fig. 1. It is also noted that TMD and TID can be expressed as TMDI with an absent of mass. Likewise, VTMD and TVMD can be VTMDI with an absent of mass. There-fore, these configurations are the major IVAs with similar components and different con-figurations. They can be formulated and modeled generically. Although the IVAs are discussed in several papers [37, 38, 39], the ESWL of structures with these IVAs are not fully addressed.”.

  1. It is suggested to discuss how two terminals of an inerter can be separated for a real structure.

Reply: Thank you for your comment. The inerter is a mechanical device that can produce a control force in proportion to the relative acceleration between the two terminals. In comparison to a mass element, which has only one terminal, an inerter produce the inertial effect with two terminals installed between two different positions of the structure. This can be implemented by different mechanical configurations, such as rack-pinon-flywheel, screw-ball systems, etc.

A sentence is added to clarify this point.

  • (Page 2, Lines 52-54) “It can produce control force in proportion to the relative acceleration at its two ends implemented by proper mechanical configurations, such as rack-pinon-flywheel, screw-ball systems, etc.”.

  1. It is suggested to enrich the introduction with some recent studies on TMD and inerter-based damper, e.g., https://doi.org/10.1016/j.jweia.2021.104836, https://doi.org/10.1016/j.engstruct.2022.115121

Reply: Thank you for your informative comment. The mentioned researches are related to inveatigation of DVAs considering nonlinear effects. In the first paper, it is TMD control of self-excited vibration for a nonlinear aeroelastic effect. The second paper is related to a DVA with cubic stiffness (a nonlinear energy sink inerter). The two papers are highly related to the manuscript, which are added as ref [40, 41]. A sentence is added to describe the investigations.

  • (Page 2, Lines 73-74) “More recently, the above DVAs are investigated considering nonlinearity effects, such as [40, 41].”.

  • Zhang, M. J.; Xu, F. Y. Tuned mass damper for self-excited vibration control: Optimization involving nonlinear aeroelastic effect. Wind. Eng. Ind. Aerodyn. 2022, 220, 104836
  • Yu, H. Y.; Zhang, M. J.; Hu, G. Effect of inerter locations on the vibration control performance of nonlinear energy sink inerter. Struct. 2022, 273, 115121

Reviewer 3 Report

IVAs are high performance novel wind-induced vibration control devices. This paper presents the equivalent static wind load (ESWL) for structures coupling with Inerter-based Vibration Absorbers (IVAs). The optimal design parameters of IVAs considering different configurations and locations are provided. Subsequently, the control performances of the IVAs with determined parameters are obtained. Finally, the corresponding ESWL are presented with gust response factor approach. This paper can be accepted considering the following comments.

(1) To my knowledge, the optimal design methods of DVAs include many approaches such as H, H2 optimizations, ect. This paper presents the Fixed-point approach, which is an analytical approximation of H optimization. Please explain the reason of adopting this method.

(2) The derivation of the solution in section 2.2 seems to be merely dependent on the coefficients of the transfer function. Is it applicable for DVAs with other configurations? Please address the application scope in the manuscript.

(3) In this paper, many IVAs are discussed, are there recommendations on the selection of the IVA based on this research?

(4) Some typos are required to be corrected.

Author Response

Response to Reviewer 3

IVAs are high performance novel wind-induced vibration control devices. This paper presents the equivalent static wind load (ESWL) for structures coupling with Inerter-based Vibration Absorbers (IVAs). The optimal design parameters of IVAs considering different configurations and locations are provided. Subsequently, the control performances of the IVAs with determined parameters are obtained. Finally, the corresponding ESWL are presented with gust response factor approach. This paper can be accepted considering the following comments.

(1) To my knowledge, the optimal design methods of DVAs include many approaches such as HH2 optimizations, ect. This paper presents the Fixed-point approach, which is an analytical approximation of H optimization. Please explain the reason of adopting this method.

Reply: Thank you for your comment. The authors agree with your comments. There are many optimization goals of the DVAs. Such as, H optimization aims at minimizing the maximum of the DAF. H2 optimization is targeted for minimizing the frequency domain integration, which corresponds to the variance of response based on the stochastic vibration theory. In the derivation, we adopted the fixed point approach for H optimal solution considering the two reasons. Firstly, the optimal results of H and H2 optimization are similar for stationary stochastic vibration responses according to previous studies. Secondly, based on the fixed point approach, the closed form solution can be derived, providing feasible formula for practical design.

In order to explain these aspects, a paragraph is added.

  • (Page 7, Lines 195-203) “It is noted that there are many optimization goals of the DVAs. Such as, H optimization aims at minimizing the maximum of the DAF. H2 optimization is targeted for minimizing the frequency domain integration, which corresponds to the variance of response based on the stochastic vibration theory. In the presented paper, we adopted the FPA for H optimal solution considering the two reasons. Firstly, the optimal results of H and H2 optimization are similar for stationary stochastic vibration responses according to previous studies [29, 39]. Secondly, based on the fixed point approach, the closed form solution can be derived, providing feasible formula for practical design.”.

(2) The derivation of the solution in section 2.2 seems to be merely dependent on the coefficients of the transfer function. Is it applicable for DVAs with other configurations? Please address the application scope in the manuscript.

Reply: Thank you for your comment. In the proposed analytical derivation with FPA in section 2.2, the results for the optimal parameters (ν and ζd∞) Eqs. (11), (12), and (15) are only based on the coefficients of the transfer function. No extra assumption is introduced. Therefore, the derivation can be applied for DVAs that follows a transfer function with a quadratic numerator polynomial and a quartic denominator polynomial. A linear DVA with a single DOF usually follow this characteristic. The abovementioned equations can be applied to such DVAs other than the IVAs investigated in this paper.

In order to address this aspect, a paragraph is added.

  • (Page 7, Lines 204-210) “Noticed that, in the proposed analytical derivation with FPA, the results for the optimal parameters (ν and ζd∞) Eqs. (11), (12), and (15) are only based on the coefficients of the transfer function. No extra assumption is introduced. Therefore, the derivation can be applied for DVAs that follows a transfer function with a quadratic numerator polynomial and a quartic denominator polynomial. A linear DVA with a single DOF usually follow this characteristic. The abovementioned equations can be applied to such DVAs other than the IVAs investigated in this paper.”.

(3) In this paper, many IVAs are discussed, are there recommendations on the selection of the IVA based on this research?

Reply: Thank you for your comment. According to the case study, VTMDI seems to be more effective in the vibration control than TMDI for it reduces more ESWL for structural design. However, it requires larger stiffness and damping to maintain an optimal design state. Therefore, the designers can make a choice according to the conclusions and methods proposed in this paper.

In order to address this aspect, several sentences are added.

  • (Page 15, Lines 388-392) “Moreover, it can be also concluded form the wind-induced vibration control analysis results that the control ratio for VTMDI is slightly lower than that for TMDI, indicating a better control performance. However, it requires larger stiffness and damping to maintain an optimal design state. Therefore, the designers can make a choice according to the conclusions and methods proposed in this paper.”.

(4) Some typos are required to be corrected.

Reply: Thank you for your comment. We have fully checked the manuscript to eliminate some typos and grammatical mistakes.

Round 2

Reviewer 1 Report

I have judged this paper to be properly revised.